# Patterns of belatacept use and risk of post-transplant lymphoproliferative disorder in US kidney transplant recipients: An analysis of the Organ Procurement and Transplantation Network database

**Wida S. Cherikh[1], Tzuyung Douglas Kou[2¤a], Julia Foutz[1], Timothy J. Baker[1], Andres Gomez-Caminero[2¤b]\***

1 Research Department, United Network for Organ Sharing, Richmond, VA, United States of America,
2 Worldwide Health Economics and Outcomes Research, Bristol Myers Squibb, Princeton, NJ, United States of America

¤a Current address: Epidemiology, Clinical Safety and Pharmacovigilance, Daiichi Sankyo US, Basking Ridge, New Jersey, United States of America
¤b Current address: Epidemiology, Merck & Co., Inc., Rahway, NJ, United States of America
\* Agc11413@yahoo.com

## Abstract

### Background

Belatacept is approved for the prophylaxis of organ rejection in Epstein-Barr virus (EBV)-seropositive kidney transplant recipients and is associated with a risk of post-transplant lymphoproliferative disorder (PTLD).

### Methods

Data from the Organ Procurement and Transplantation Network were used to examine patterns of belatacept use, describe patient characteristics, and estimate risk of PTLD in EBV-seropositive, kidney-only transplant recipients receiving belatacept- or calcineurin inhibitor (CNI)-based immunosuppression as part of US Food and Drug Administration-mandated safety monitoring.

### Results

During the study period (June 15, 2011–June 14, 2016), 94.9% (1631/1719) of belatacept-treated and 89.7% (59,992/66,905) of CNI-treated patients with known EBV serostatus were EBV seropositive. Among EBV-seropositive patients, 50.2% (belatacept) and 56.8% (CNI) received a standard criteria donor kidney, 59.5% and 18.7% received basiliximab induction, and 22.9% and 50.8% received antithymocyte globulin induction. PTLD developed in nine belatacept-treated patients (two with central nervous system [CNS] involvement) and 225 CNI-treated patients (nine with CNS involvement). Four and 81 patients, respectively, died due to PTLD. Kaplan–Meier analysis did not show a significant between-

**Data Availability Statement:** The data underlying the results presented in the study are available upon request from the Organ Procurement and Transplantation Network (OPTN) by submitting the request at https://optn.transplant.hrsa.gov/data/view-data-reports/request-data. Individual researchers requesting OPTN datasets must sign a Data Use Agreement and will receive the complete set of raw datasets used for the current analysis.

**Funding:** This study was financially supported by Bristol Myers Squibb (BMS) [https://www.bms.com] in the form of an award through the funding of medical writing and editorial assistance. BMS participated in the study design, data curation and analysis, decision to publish, preparation of the manuscript, medical writing, and editorial assistance for development of the manuscript. AG-C was a salaried employee of Bristol Myers Squibb at the time of the study. The specific roles of this author are articulated in the 'author contributions' section. This work was also financially supported by Health Resources and Services Administration in the form of a contract award (234-2005-370011C). The content of this publication is the responsibility of the authors alone and does not necessarily reflect the views or policies of the Department of Health and Human Services, nor does mention of trade names, commercial products, or organizations imply endorsement by the U.S. Government. The funders had no additional undeclared role in study design, data collection and analysis, decision to publish, or preparation of the manuscript.

**Competing interests:** W.S.C., J.F., and T.J.B. have served as consultants to and have received research support from Bristol Myers Squibb. T.D.K. was a salaried employee of Bristol Myers Squibb at the time of the study and is currently an employee of Daiichi Sankyo US. A.G.-C. was a salaried employee of Bristol Myers Squibb at the time of the study and is currently an employee of Merck & Co., Inc. This does not alter our adherence to PLOS ONE policies on sharing data and materials.

group difference in PTLD estimated incidence rates within 5 years (0.70% versus 0.48%, respectively; p = 0.18). Additionally, estimated PTLD incidence was not significantly different between treatment groups in a propensity score matched cohort.

## Conclusions

The majority of adult kidney-only transplant recipients treated with belatacept in routine clinical practice are EBV seropositive. In this study, the risk of PTLD in these patients, while higher than for CNI-based immunosuppression, remained low after adjusting for differences in patient characteristics.

## Trial registration

These studies are registered at ClinicalTrials.gov: NCT01670058 and NCT01656343.

## Introduction

Although the calcineurin inhibitors (CNIs) such as tacrolimus and cyclosporine have reduced acute rejection rates and improved shorter term clinical outcomes in kidney transplant recipients [1], CNI-based immunosuppressive regimens are associated with nephrotoxicity and allograft injury potentially leading to graft loss, cardiovascular risk, neurotoxicity, and islet cell toxicity [2–4]. Belatacept is a fusion protein composed of the Fc fragment of human IgG1 linked to the extracellular domain of modified cytotoxic T-lymphocyte-associated antigen 4 that selectively inhibits T-cell activation through blockade of the CD28/CD80-86 costimulatory pathway [5–7]. In two phase 3 clinical trials (Belatacept Evaluation of Nephroprotection and Efficacy as First-Line Immunosuppression Trial [BENEFIT] and BENEFIT-Extended Criteria Donors [BENEFIT-EXT]), kidney-only transplant recipients administered belatacept-based immunosuppression had comparable patient/graft survival and superior kidney function versus those administered cyclosporine-based immunosuppression [8–10]. Relative to cyclosporine-treated patients, belatacept-treated patients have been shown to experience long-term (7 years) benefit in kidney function and a reduction in the risk of graft loss or death [11].

Post-transplant lymphoproliferative disorder (PTLD) is a severe but infrequent complication of transplantation, reported in 0.6%–3.2% of kidney transplant recipients [12–18]. PTLD can develop within 1 year post-transplant or have a late onset (2 to >5 years post-transplant). Approximately 55%–65% of patients with PTLD are positive for Epstein-Barr virus (EBV), and the risk of PTLD is increased when an EBV-seronegative patient receives a kidney from an EBV-seropositive donor [19]. In the BENEFIT and BENEFIT-EXT studies, the rate of PTLD was approximately 9-fold higher among belatacept-treated patients who were EBV seronegative or whose serostatus was unknown (5.8% [8/139]) than among those who were EBV seropositive (0.6% [5/810]) [8–11]. In 2011, belatacept was approved in the United States (US) and European Union for the prophylaxis of organ rejection in EBV-seropositive kidney transplant recipients [20, 21].

Post-marketing studies are used to evaluate patients in a routine clinical care setting and to further describe real-world treatment patterns and evaluate safety [22–24]. As part of US Food and Drug Administration (FDA)-mandated safety monitoring requirement after approval, two retrospective observational epidemiology studies (IM103-074 and IM103-075) of belatacept were undertaken in the US. These studies used records from the Organ Procurement and

Transplantation Network (OPTN) database, which is currently managed by the United Network for Organ Sharing. Here, we present the results of these observational studies, including patterns of belatacept use and incidence rates of PTLD in adult, EBV-seropositive, kidney-only transplant recipients.

## Patients and methods

### Patient population

The retrospective studies IM103-074 (NCT01670058) and IM103-075 (NCT01656343) used data from the OPTN database as of April 12, 2019, specifically those from transplant candidate registration, transplant recipient registration (TRR), transplant recipient follow-up (TRF), and donor records.

All adult kidney-only transplant recipients in the US who received belatacept or CNI (tacrolimus or cyclosporine) at time of transplant between June 15, 2011, and June 14, 2016, were included. Based on information on the TRR records, patients were categorized as having received either belatacept or CNI. An intent-to-treat approach was used in the analyses (i.e., any report of belatacept or CNI use on the TRR records, regardless of any switches that may have occurred thereafter), and individuals who received both belatacept and CNI were included in the belatacept treatment group.

Patient follow-up data were submitted to the OPTN database at 6 months after transplant and then annually thereafter until the end of the study observation period or the occurrence of death, graft loss, or loss to follow-up, whichever came first. These studies were conducted in accordance with the International Society for Pharmacoepidemiology Guidelines for Good Epidemiology Practices [25]. As the current study utilized limited datasets containing patient-level, non-identifiable data extracted from the OPTN research database, it was determined to be non-human subject research and, therefore, IRB-exempt by the Chesapeake Institutional Review Board. Authors did not have access to information that could identify individual participants during or after data analysis.

### Study objectives

The primary objectives of the IM103-074 study were to examine patterns of belatacept use and to describe the demographic and clinical characteristics of adult, EBV-seropositive, kidney-only transplant recipients treated with belatacept- or CNI-based regimens from the time of transplant. The primary objectives of the IM103-075 study were to estimate the incidence rates of PTLD in EBV-seropositive, kidney-only transplant recipients from the time of transplant and to compare PTLD incidence rates between those patients who received belatacept- versus CNI-based immunosuppression.

### Statistics

The associations between treatment (belatacept or CNI) and age at transplant, sex, body weight, body mass index (BMI), donor type, donor EBV serostatus, donor-to-recipient cytomegalovirus (CMV) serostatus, use of adjunctive treatment at transplant, or type of antibody induction were assessed using chi-square or Fisher's exact tests (as appropriate) for categorical variables and the Wilcoxon rank sum test for continuous variables. For categorical variables, the number of missing observations was presented but was not included in the reported distributions. The p-values were calculated using the Pearson chi-square test for categorical variables and Wilcoxon rank sum test for continuous variables.

Incidence rates of PTLD and 95% confidence intervals (CIs) were estimated as the number of new PTLD cases during the study period per 1000 patient-years at risk. Years at risk were computed as the difference (in days) between the transplant date (on TRR record) and the date of PTLD diagnosis (on Post-Transplant Malignancy record) for those who developed PTLD or between the transplant date and the most recent patient status date (on TRF record) for those who did not develop PTLD, divided by 365.25. The cumulative incidence rates of PTLD in each treatment group and the corresponding 95% CIs were estimated. The Kaplan–Meier method was also used to compare the risk of PTLD between treatment groups.

Propensity score matching (PSM) multivariable analysis was performed to increase the comparability of belatacept and CNI groups and adjust for potential bias and confounding due to any difference in patient characteristics between treatment groups. Covariates in the propensity score model included age at transplant, sex, BMI, history of a prior transplant, use of mycophenolate (mycophenolate mofetil [CellCept] or sodium mycophenolate [Myfortic]) at transplant, type of induction therapy, and steroid use at transplant. Other covariates included donor type (i.e., living versus standard or expanded criteria deceased [deceased donor aged >60 years or deceased donor aged 50–59 years with ≥2 of the following: cerebrovascular cause of death, terminal serum creatinine >1.5 mg/dL, history of hypertension]), donor EBV serostatus, and donor-to-recipient CMV serostatus. Missing data were imputed using the median or most frequent category, and the propensity score distribution for belatacept and CNI groups was evaluated for disjoint ranges. The greedy algorithm was used to create a propensity score matched-pair sample. A 1:1 matched case:control cohort was created. Differences in covariates between matched pairs were evaluated using the Pearson chi-square test for categorical variables and Kruskal-Wallis test for continuous variables. The incidence of PTLD per 1000 patient-years and Kaplan–Meier incidence of PTLD were estimated in the matched cohort. Additionally, Cox proportional hazards modeling of time to PTLD was conducted for the matched cohort.

Statistical analyses were performed using SAS version 9.4 (SAS Institute, Inc., Cary, NC, USA), with p-values <0.05 considered significant.

## Results

### Patients

Between June 15, 2011, and June 14, 2016, 1737 kidney-only transplant recipients were identified as having received belatacept, and 74,637 received CNI at transplant (Fig 1). A total of 798 patients were identified as having received both belatacept and CNI at transplant and were included in the belatacept treatment group. The median duration of follow-up was 1389 days in the belatacept group and 1436 days in CNI group.

EBV serostatus was known in 1719 and 66,905 patients in the belatacept and CNI treatment groups, respectively. Of the 1719 belatacept-treated patients with known EBV serostatus, 1631 (94.9%) were EBV seropositive and 88 (5.1%) were EBV seronegative. Of the 66,905 CNI-treated patients with known EBV serostatus, 59,992 (89.7%) were EBV seropositive and 6913 (10.3%) were EBV seronegative.

Among EBV-seropositive recipients, baseline demographic and clinical characteristics were comparable between the two treatment groups (Table 1). The mean age was ~52 years, and the mean BMI was ~28 kg/m$^2$. Most patients were male (61%) and alive at last assessment (89%). Nearly all (98%) patients received kidneys from EBV-seropositive donors, and ~42% had a positive-positive donor-recipient CMV serostatus combination at transplant. The proportion of patients treated with a mechanistic target of rapamycin (mTOR) inhibitor at transplant was low (~2%).

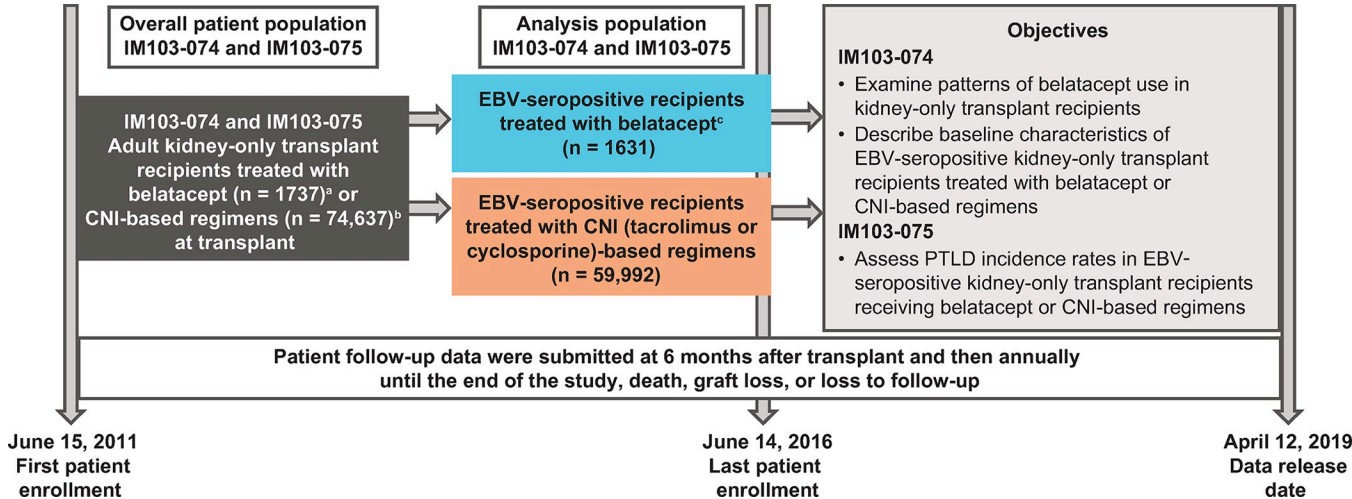

**Fig 1. Cohort studies using data captured in the OPTN database.** [a]In total, 1719 patients had known EBV serostatus; of these, 1631 (94.9%) were EBV seropositive. [b]In total, 66,905 patients had known EBV serostatus; of these, 59,992 (89.7%) were EBV seropositive. [c]A total of 798 patients treated with belatacept also received CNI at transplant. CNI, calcineurin inhibitor; EBV, Epstein-Barr virus; OPTN, Organ Procurement and Transplantation Network; PTLD, post-transplant lymphoproliferative disorder.

## PTLD events and PTLD incidence rates

Among EBV-seropositive recipients, nine PTLD events were reported up to 5 years post-transplant in the belatacept treatment group and 204 were reported in the CNI treatment group. Among the 1631 patients in the belatacept treatment group, there were three cases of PTLD within 1 year post-transplant, six within 2 years, eight within 3 and 4 years, and nine within 5 years (Table 2). The corresponding values among the 59,992 patients in the CNI treatment group were 44, 109, 146, 178, and 204.

As shown in Fig 2, the Kaplan–Meier incidence rates of PTLD increased from year 1 to year 5 post-transplant in both treatment groups. At year 1, the PTLD incidence rate was 0.19% (95% CI 0.06−0.58) in the belatacept treatment group and 0.08% (95% CI 0.06−0.10) in the CNI treatment group. At year 5, these rates increased to 0.70% (95% CI 0.35−1.42) and 0.48% (95% CI 0.41−0.55), respectively. The incidence rate of PTLD within 5 years post-transplant did not differ significantly between treatment groups (log-rank p = 0.18). Because of the small number of events among belatacept-treated patients (nine by year 5), no further statistical comparison of PTLD incidence rates between treatment groups was undertaken, and Kaplan–Meier plots were not constructed.

The cumulative incidence rates of PTLD per 1000 patient-years at risk between year 1 and year 5 post-transplant for each treatment group are shown in Table 2. At 5 years post-transplant, the PTLD incidence rate was 7.80 per 1000 patient-years in the belatacept treatment group and 4.66 per 1000 patient-years in the CNI treatment group.

## Characteristics of patients who developed PTLD

Baseline demographic and clinical characteristics of the nine belatacept-treated and 225 CNI-treated patients who developed PTLD as of the time of the analysis are presented in Table 3. No significant differences between treatment groups were seen. Of note, three of the nine belatacept-treated patients who developed PTLD also received CNI at transplant.

PTLD histopathology was reported in 88.9% of belatacept-treated and 89.3% of CNI-treated patients who developed PTLD. In general, histopathology subtypes showed broad and similar

**Table 1. Demographic and clinical characteristics of adult, EBV-seropositive kidney transplant recipients by treatment group.**

| Characteristics | Belatacept[a] (n = 1631) | CNI (n = 59,992) | p-value[b] |
|---|---|---|---|
| Age at transplant, years | | | 0.84 |
| Mean (SD) | 51.9 (13.7) | 51.8 (13.6) | |
| Median (IQR) | 53.0 (42.0–63.0) | 53.0 (42.0–62.0) | |
| Sex, n (%) | | | 0.13 |
| Female | 613 (37.6) | 23,673 (39.5) | |
| Male | 1018 (62.4) | 36,319 (60.5) | |
| Body weight, kg | | | 0.29 |
| Mean (SD) | 82.3 (18.9) | 81.8 (19.4) | |
| Median (IQR) | 81.0 (68.0–95.3) | 80.5 (67.9–94.5) | |
| Missing | 1 | 195 | |
| BMI, kg/m$^2$ | | | 0.14 |
| Mean (SD) | 27.8 (5.2) | 28.1 (5.5) | |
| Median (IQR) | 27.5 (23.8–31.9) | 27.7 (24.1–31.8) | |
| Missing | 17 | 399 | |
| Previous transplant, n (%) | | | <0.0001 |
| No | 1489 (91.3) | 52,348 (87.3) | |
| Yes | 142 (8.7) | 7644 (12.7) | |
| Donor type, n (%) | | | <0.0001 |
| Expanded criteria donor[c] | 190 (11.6) | 5747 (9.6) | |
| Standard criteria donor | 819 (50.2) | 34,096 (56.8) | |
| Living donor | 622 (38.1) | 20,149 (33.6) | |
| Donor EBV serostatus, n (%) | | | 0.98 |
| Negative | 29 (1.9) | 1062 (1.9) | |
| Positive | 1486 (98.1) | 54,717 (98.1) | |
| Missing[d] | 116 | 4213 | |
| Donor-recipient CMV serostatus, n (%) | | | 0.31 |
| Negative-negative | 280 (18.1) | 9751 (16.5) | |
| Negative-positive | 371 (24.0) | 14,766 (25.0) | |
| Positive-negative | 249 (16.1) | 9993 (16.9) | |
| Positive-positive | 648 (41.9) | 24,567 (41.6) | |
| Missing[d,e] | 83 | 915 | |
| Type of antibody induction, n (%) | | | <0.0001 |
| Alemtuzumab | 155 (9.5) | 9404 (15.7) | |
| Basiliximab | 971 (59.5) | 11,213 (18.7) | |
| Antithymocyte globulin[f] | 373 (22.9) | 30,454 (50.8) | |
| Other/multiple drugs[g] | 60 (3.7) | 2885 (4.8) | |
| No induction | 72 (4.4) | 6036 (10.1) | |
| Mycophenolate adjunctive treatment, n (%) | | | 0.0005 |
| No | 88 (5.4) | 2242 (3.7) | |
| Yes | 1543 (94.6) | 57,750 (96.3) | |
| Steroid use at transplant, n (%) | | | <0.0001 |
| No | 317 (19.4) | 18,667 (31.1) | |
| Yes | 1314 (80.6) | 41,325 (68.9) | |
| mTOR inhibitor use at transplant, n (%) | | | 0.91 |
| No | 1594 (97.7) | 58,656 (97.8) | |
| Yes | 37 (2.3) | 1336 (2.2) | |
| Acute rejection rate at 1 year, n (%) | | | <0.0001 |

*(Continued)*

**Table 1.** (Continued)

| Characteristics | Belatacept[a] (n = 1631) | CNI (n = 59,992) | p-value[b] |
|---|---|---|---|
| No | 1139 (77.7) | 48,292 (90.5) | |
| Yes | 326 (22.3) | 5091 (9.5) | |
| Missing[d] | 166 | 6609 | |
| Most recent patient status, n (%) | | | 0.89 |
| Alive | 1453 (89.1) | 53,378 (89.0) | |
| Dead | 178 (10.9) | 6614 (11.0) | |

[a]A total of 798 transplant recipients received both belatacept and CNI at transplant.

[b]Calculated via the Pearson chi-square test for categorical variables and Wilcoxon rank sum test for continuous variables.

[c]Expanded criteria donor is defined as a deceased donor aged >60 years or aged 50–59 years with ≥2 of the following: cerebrovascular cause of death, terminal serum creatinine >1.5 mg/dL, or history of hypertension.

[d]Missing patients were excluded from percentage calculations.

[e]Included negative-unknown, positive-unknown, unknown-negative, unknown-positive, and unknown-unknown donor-recipient CMV serostatus.

[f]Refers to rabbit-derived antithymocyte globulin (Thymoglobulin).

[g]Included equine-derived antithymocyte globulin (Atgam), muromonab-CD3, daclizumab, or multiple drugs.

BMI, body mass index; CMV, cytomegalovirus; CNI, calcineurin inhibitor; EBV, Epstein-Barr virus; IQR, interquartile range; mTOR, mechanistic target of rapamycin; SD, standard deviation.

distributions in both treatment groups. The predominant histopathologic features in the CNI treatment group were monomorphic (34.8%) and polymorphic (17.9%) PTLD (lymphoma) followed by multiple myeloma/plasmacytoma (10.9%). Of the four belatacept-treated patients who developed PTLD with a known anatomic location, two (50.0%) had CNS PTLD and two (50.0%) had non-CNS PTLD. Of the 25 CNI-treated patients who developed PTLD with a known anatomic location, nine (36.0%) had CNS PTLD and 16 (64.0%) had non-CNS PTLD. PTLD was the cause of death in 44.4% of belatacept-treated and 36.0% of CNI-treated patients who developed PTLD. Of note, one case of PTLD was reported in an EBV-seronegative patient who received belatacept at transplant; this patient died due to PTLD within 1 year of transplant.

## PSM multivariable analyses

PSM was used to further describe the risk of PTLD in belatacept-treated patients after adjusting for any differences in baseline characteristics between the two treatment groups with a 1:1 matching scheme. Characteristics used for matching are shown in S1 Table. A total of 1631

**Table 2. Cumulative incidence rates of PTLD per 1000 patient-years at risk within 5 years post-transplant in adult, EBV-seropositive kidney transplant recipients by treatment group.**

| | Belatacept (n = 1631) | | | CNI (n = 59,992) | | |
|---|---|---|---|---|---|---|
| Year | PTLD cases, n | Total patient-years | PTLD rate[a] (95% CI) | PTLD cases, n | Total patient-years | PTLD rate[a] (95% CI) |
| 1 | 3 | 1600.2 | 1.87 (0.39–5.48) | 44 | 58,674.2 | 0.75 (0.54–1.01) |
| 2 | 6 | 1560.2 | 3.85 (1.41–8.37) | 109 | 57,198.9 | 1.91 (1.56–2.30) |
| 3 | 8 | 1460.8 | 5.48 (2.36–10.80) | 146 | 54,200.1 | 2.69 (2.27–3.18) |
| 4 | 8 | 1310.5 | 6.11 (2.64–12.03) | 178 | 49,190.8 | 3.62 (3.11–4.19) |
| 5 | 9 | 1154.4 | 7.80 (3.57–14.80) | 204 | 43,811.5 | 4.66 (4.04–5.34) |

[a]Per 1000 patient-years at risk.

CI, confidence interval; CNI, calcineurin inhibitor; EBV, Epstein-Barr virus; PTLD, post-transplant lymphoproliferative disorder.

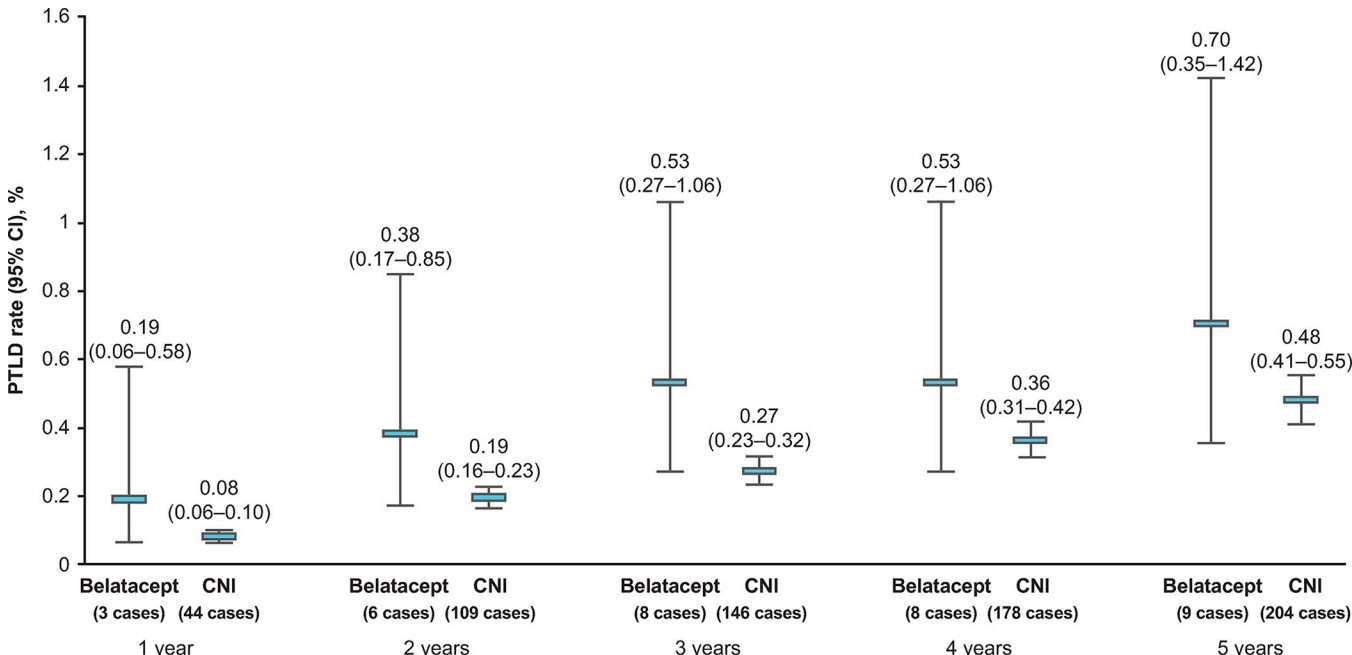

**Fig 2. Kaplan–Meier incidence rates of PTLD within 1–5 years post-transplant.** [a]Three of the patients who developed PTLD had received both belatacept and CNI at transplant. CI, confidence interval; CNI, calcineurin inhibitor; PTLD, post-transplant lymphoproliferative disorder.

patients from the CNI treatment group were matched to all 1631 EBV-seropositive, kidney-only transplant recipients in the belatacept group. The matched cohort and propensity scores balance was analyzed using the standardized mean differences of the covariates used for matching. The analysis revealed that all variables were well balanced in the two treatment groups (S1 Fig), and that the distributions of the matched and overall propensity scores between patients in the belatacept and CNI treatment groups were similar (S2 Fig). As a last assessment of covariate balance, the matched variables were compared between the two treatment groups, and no significant differences were observed in any of the matching variables (S1 Table).

In the matched cohort, the Kaplan–Meier incidence rate of PTLD within 5 years post-transplant did not differ significantly between treatment groups (log-rank p = 0.27); by 5 years post-transplant, the incidence of PTLD was 0.70% in the belatacept group and 0.37% in the CNI group (Table 4). The incidence of PTLD per 1000 patient-years at risk was higher at each yearly post-transplant time point for patients in the belatacept versus CNI treatment group, but the differences did not reach statistical significance in that the 95% CIs overlapped (Table 5).

A Cox proportional hazards model of the time to development of PTLD was performed on the matched cohort, with treatment group (belatacept versus CNI) as the only risk factor. The hazard ratio for PTLD was 1.84 (95% CI 0.62–5.50), indicating that the risk of PTLD was higher in belatacept-treated compared with CNI-treated patients; however, the difference in risk of PTLD between treatment groups did not reach statistical significance (p = 0.27).

## Discussion

This analysis represents a real-world examination of EBV-seropositive kidney-only, transplant recipients who received belatacept as part of routine clinical practice and was conducted as

**Table 3. Demographic and clinical characteristics of adult, EBV-seropositive kidney transplant recipients who developed PTLD by treatment group.**

| Characteristics | Belatacept[a] (n = 9) | CNI (n = 225) |
|---|---|---|
| Age at transplant, years | | |
| Mean (SD) | 60.6 (11.8) | 57.3 (13.5) |
| Median (IQR) | 61.0 (55.0–68.0) | 60.0 (50.0–67.0) |
| Sex, n (%) | | |
| Female | 1 (11.1) | 65 (28.9) |
| Male | 8 (88.9) | 160 (71.1) |
| Body weight (kg) | | |
| Mean (SD) | 77.5 (26.9) | 81.8 (19.1) |
| Median (IQR) | 63.7 (62.0–80.9) | 80.7 (69.4–92.5) |
| Missing | 0 | 1 |
| BMI (kg/m$^2$) | | |
| Mean (SD) | 25.5 (6.4) | 27.7 (5.8) |
| Median (IQR) | 24.5 (21.3–25.6) | 27.3 (23.9–31.3) |
| Missing | 0 | 1 |
| Previous transplant, n (%) | | |
| No | 8 (88.9) | 200 (88.9) |
| Yes | 1 (11.1) | 25 (11.1) |
| Donor type, n (%) | | |
| Expanded criteria donor | 1 (11.1) | 36 (16.0) |
| Standard criteria donor | 7 (77.8) | 104 (46.2) |
| Living donor | 1 (11.1) | 85 (37.8) |
| Donor EBV serostatus, n (%) | | |
| Negative | 1 (12.5) | 4 (1.9) |
| Positive | 7 (87.5) | 203 (98.1) |
| Missing[b] | 1 | 18 |
| Donor-recipient CMV serostatus, n (%) | | |
| Negative-negative | 2 (22.2) | 34 (15.5) |
| Negative-positive | 2 (22.2) | 53 (24.2) |
| Positive-negative | 1 (11.1) | 45 (20.5) |
| Positive-positive | 4 (44.4) | 87 (39.7) |
| Missing[b,c] | 0 | 6 |
| Type of antibody induction, n (%) | | |
| Alemtuzumab | 1 (11.1) | 47 (20.9) |
| Basiliximab | 2 (22.2) | 47 (20.9) |
| Antithymocyte globulin[d] | 4 (44.4) | 107 (47.6) |
| Other/multiple drugs[e] | 0 | 8 (3.6) |
| No induction | 2 (22.2) | 16 (7.1) |
| Mycophenolate adjunctive treatment, n (%) | | |
| No | 0 | 7 (3.1) |
| Yes | 9 (100) | 218 (96.9) |
| Steroid use at transplant, n (%) | | |
| No | 4 (44.4) | 81 (36.0) |
| Yes | 5 (55.6) | 144 (64.0) |
| mTOR inhibitor use at transplant, n (%) | | |
| No | 9 (100.0) | 219 (97.3) |
| Yes | 0 | 6 (2.7) |

*(Continued)*

**Table 3.** (Continued)

| Characteristics | Belatacept[a] (n = 9) | CNI (n = 225) |
|---|---|---|
| Acute rejection rate at 1 year, n (%) | | |
| No | 6 (85.7) | 177 (86.8) |
| Yes | 1 (14.3) | 27 (13.2) |
| Missing[b] | 2 | 21 |
| Location of PTLD, n (%) | | |
| CNS | 2 (50.0) | 9 (36.0) |
| Non-CNS | 2 (50.0) | 16 (64.0) |
| Not reported[b] | 5 | 200 |
| PTLD pathology, n (%) | | |
| Polymorphic hyperplasia | 0 | 1 (0.5) |
| Polymorphic PTLD (lymphoma) | 2 (25.0) | 36 (17.9) |
| Monomorphic PTLD (lymphoma) | 2 (25.0) | 70 (34.8) |
| Multiple myeloma/plasmacytoma | 1 (12.5) | 22 (10.9) |
| Hodgkin's disease | 1 (12.5) | 6 (3.0) |
| Other | 2 (25.0) | 66 (32.8) |
| Missing[b] | 1 | 24 |
| Patient status, n (%) | | |
| Alive | 4 (44.4) | 126 (56.0) |
| Deceased of PTLD | 4 (44.4) | 81 (36.0) |
| Deceased of other causes | 1 (11.1) | 16 (7.1) |
| Lost to follow-up | 0 | 2 (0.9) |

[a]Three transplant recipients from three different centers received both belatacept and CNI at transplant.

[b]Missing/not reported patients were excluded from percentage calculations.

[c]Included negative-unknown, positive-unknown, unknown-negative, unknown-positive, and unknown-unknown donor-recipient CMV serostatus.

[d]Refers to rabbit-derived antithymocyte globulin.

[e]Included patients treated with equine-derived antithymocyte globulin (n = 1), rabbit-derived antithymocyte globulin and basiliximab (n = 6), or alemtuzumab and basiliximab (n = 1).

BMI, body mass index; CMV, cytomegalovirus; CNI, calcineurin inhibitor; CNS, central nervous system; EBV, Epstein-Barr virus; IQR, interquartile range; mTOR, mechanistic target of rapamycin; PTLD, post-transplant lymphoproliferative disorder; SD, standard deviation.

part of US FDA-mandated post-marketing surveillance obligations. Most belatacept-treated patients captured in the OPTN database were EBV seropositive (94.9%) and received concomitant immunosuppressive drugs consistent with the approved labeling for the use of belatacept [26], specifically mycophenolate (94.6%) and corticosteroids (80.6%) for maintenance, and basiliximab for induction (59.5%). Given the approved labeling, it was anticipated that fewer patients in the belatacept treatment group would receive antithymocyte globulin for immunosuppressive induction as compared with the CNI treatment group (22.9% versus 50.8%, respectively).

The demographic and clinical characteristics of patients in this analysis differed somewhat from those who participated in the phase 3 BENEFIT and BENEFIT-EXT clinical trials. Belatacept-treated patients in the real-world tended to be older than those receiving the approved dosing regimen in BENEFIT (mean age of 51.9 versus 42.6 years) [10] and younger than those receiving the approved dosing regimen in BENEFIT-EXT (mean age of 56.1 years) [9]. In terms of kidney donor type and quality, two of the covariates affecting the risk of failure,

**Table 4. Kaplan–Meier incidence rates of PTLD in the matched cohort.**

| Year | No. treatments | Belatacept (n = 1631) | | | CNI (n = 1631) | | |
|---|---|---|---|---|---|---|---|
| | | PTLD cases, n | % of PTLD cases | 95% CI of PTLD rate | PTLD cases, n | % of PTLD cases | 95% CI of PTLD rate |
| 1 | 1631 | 3 | 0.19 | 0.06−0.58 | 2 | 0.13 | 0.03−0.50 |
| 2 | 1631 | 6 | 0.38 | 0.17−0.85 | 4 | 0.26 | 0.10−0.68 |
| 3 | 1631 | 8 | 0.53 | 0.27−1.06 | 4 | 0.26 | 0.10−0.68 |
| 4 | 1631 | 8 | 0.53 | 0.27−1.06 | 5 | 0.37 | 0.15−0.93 |
| 5 | 1631 | 9 | 0.70 | 0.35−1.42 | 5 | 0.37 | 0.15−0.93 |

CI, confidence interval; CNI, calcineurin inhibitor; PTLD, post-transplant lymphoproliferative disorder.

11.6% of patients in the real-world setting received an extended criteria deceased donor kidney, while none of the BENEFIT participants did. All of the transplanted and treated participants in BENEFIT-EXT received kidneys at higher risk of subsequent graft failure, either because they came from an extended criteria deceased donor (71%), were donated following cardiac death (10%), or had estimated cold ischemia (preservation) times of 24 hours or longer (42%) with or without one of the other two risk factors [9, 10]. Additionally, relative to kidney transplant recipients receiving the approved dosing regimen in BENEFIT [10], fewer belatacept-treated patients in the OPTN database had not undergone prior transplant (91% versus 97%).

Among EBV-seropositive, kidney-only transplant recipients in the present analysis, the 5-year Kaplan–Meier incidence rate of PTLD was low, regardless of whether patients received belatacept or CNI at transplant (0.70% and 0.48%, respectively). Although the incidence rate of PTLD over the first 5 years post-transplant was numerically higher among belatacept- versus CNI-treated patients, the difference was not statistically significant. However, it is possible that this observed difference between treatment groups is a consequence of clinical confounding factors. As noted above (and in accordance with the label), proportionally more patients in the belatacept treatment group were receiving corticosteroids; therefore, kidney transplant recipients receiving belatacept may have experienced a greater level of immunosuppression, which may have reduced the effectiveness of immune surveillance for malignancy. In addition, the higher rate of acute rejection at 1 year post-transplant in the belatacept treatment group may have contributed to the short-term intensification of overall immunosuppression required for treatment and control of rejection. Regardless, the 1-year PTLD incidence rates among belatacept-treated patients in the present analysis are consistent with those reported at 7 years post-transplant in EBV-seropositive patients receiving the approved dosing regimen for belatacept in BENEFIT (0.1%–0.2%) [11] and BENEFIT-EXT (0.1%–0.25%) [8], and the

**Table 5. Cumulative incidence rates of PTLD per 1000 patient-years at risk in the matched cohort.**

| Year | Belatacept (n = 1631) | | | CNI (n = 1631) | | |
|---|---|---|---|---|---|---|
| | PTLD cases, n | Total patient-years | PTLD rate[a] (95% CI) | PTLD cases, n | Total patient-years | PTLD rate[a] (95% CI) |
| 1 | 3 | 1600.2 | 1.87 (0.39–5.48) | 2 | 1596.9 | 1.25 (0.15–4.52) |
| 2 | 6 | 1560.2 | 3.85 (1.41–8.37) | 4 | 1559.4 | 2.57 (0.70–6.57) |
| 3 | 8 | 1460.8 | 5.48 (2.36–10.79) | 4 | 1481.2 | 2.70 (0.74–6.91) |
| 4 | 8 | 1310.5 | 6.10 (2.64–12.03) | 5 | 1347.9 | 3.71 (1.20–8.66) |
| 5 | 9 | 1154.4 | 7.80 (3.56–14.80) | 5 | 1203.1 | 4.16 (1.35–9.70) |

[a]Per 1000 patient-years at risk.

CI, confidence interval; CNI, calcineurin inhibitor; PTLD, post-transplant lymphoproliferative disorder.

5-year PTLD incidence rate was within the range of those reported in the literature (0.6%–3.2%) [12–18].

Of the nine cases of PTLD reported among belatacept-treated patients, all but one had occurred by year 3 (88.9%) post-transplant. At 3 years post-transplant, 71.6% (146/204) of all cases of PTLD observed among CNI-treated patients had occurred. PTLD tended to occur in older patients; the mean age of belatacept- and CNI-treated patients who developed PTLD was 60.6 and 57.3 years, respectively. Because of the small number of events among belatacept-treated patients (n = 9), subgroup analyses of PTLD incidence rates (vis-à-vis demographic or clinical characteristics) were not performed, but to account for differences in baseline covariates, a PSM multivariable analysis was undertaken. Data from the matched cohort confirmed the findings of the overall study population; there were no statistically significant differences between risks of PTLD in belatacept- and CNI-treated patients, regardless of whether PTLD incidence rates were calculated based on patient-years at risk or using the Kaplan–Meier method.

The limitations of this analysis are largely those inherent to a retrospective observational study, and these should be considered when interpreting our findings. As expected, patient selection is less stringent in real-world clinical practice versus that which occurs in clinical studies. For example, ~5% of patients receiving belatacept in our analysis were EBV seronegative, even though belatacept is indicated solely for use in patients who are EBV seropositive [27]. Although transplant hospitals are required to submit complete and accurate data to the OPTN, they have the option to indicate when data are not available or unknown. Additionally, during the analysis period, submission of certain data elements was not required. For example, the anatomic location of PTLD became an optional data element for reporting purposes as of February 7, 2007, and ceased to be collected at all as of March 31, 2015. As a consequence, the anatomic location(s) at which PTLD developed was missing for most patients in our analysis. Also, some variables that could potentially impact the risk of PTLD are not specifically captured in the OPTN database, such as dosing frequency, details of switchover from CNI to belatacept, or immunosuppression usage beyond 1 year; data collection of comorbidities at transplant is also limited. Despite these caveats, the OPTN database was selected for this analysis because it captures all transplants that have been performed in the US since October 1, 1987. Third, overall use of belatacept-based immunosuppression in *de novo* renal transplantation was found to be limited, and the low number of PTLD events in both the belatacept and CNI groups (nine and 225 cases, respectively) limited the types of statistical analyses that could be performed; results should be interpreted with caution. Fourth, some degree of reporting bias may have occurred, including the underreporting of PTLD in the OPTN database. Finally, our study was limited to PTLD risk within 5 years of transplant. Future studies would potentially be needed to assess the longer-term risk of PTLD from belatacept usage.

In summary, the results of these real-world observational studies indicate that most adult kidney-only transplant recipients prescribed belatacept in routine clinical practice are EBV seropositive and that belatacept is administered largely in accordance with the approved labeling. Among adult, EBV-seropositive, kidney-only transplant recipients receiving belatacept included in this study, the risk of PTLD remained relatively low, that is, comparable to that observed among patients receiving CNI-based immunosuppression and similar to that in the belatacept clinical registration trials.

## Supporting information

**S1 Fig. Standardized mean differences of covariates used to match belatacept and CNI at transplant in the matched cohort.** [a]Using imputed value for missing. [b]Refers to rabbit-

derived antithymocyte globulin (Thymoglobulin). [c]"Region" refers to observations with over-lapping propensity scores for the treated (belatacept) and control (CNI) groups. BMI, body mass index; CMV, cytomegalovirus; CNI, calcineurin inhibitor; EBV, Epstein-Barr virus. (TIF)

**S2 Fig. Box plots of propensity scores for covariates at baseline.** [a]"Region" refers to observations with overlapping propensity scores for the treated (belatacept) and control (CNI) groups. (TIF)

**S1 Table. Demographic and clinical characteristics in the matched cohort.** (PDF)

# Acknowledgments

We would like to acknowledge Samantha Noreen, PhD, of United Network for Organ Sharing for her assistance with PSM analyses. Medical writing and editorial assistance were provided by Vasupradha Vethantham, PhD, Kakoli Parai, PhD, and Jennifer Robertson, PhD, of Ashfield MedComms, an Inizio company.

# Author Contributions

**Conceptualization:** Tzuyung Douglas Kou, Andres Gomez-Caminero.

**Data curation:** Wida S. Cherikh, Tzuyung Douglas Kou, Julia Foutz, Timothy J. Baker, Andres Gomez-Caminero.

**Formal analysis:** Wida S. Cherikh, Tzuyung Douglas Kou, Julia Foutz, Timothy J. Baker, Andres Gomez-Caminero.

**Investigation:** Wida S. Cherikh, Tzuyung Douglas Kou, Julia Foutz, Timothy J. Baker, Andres Gomez-Caminero.

**Writing – original draft:** Wida S. Cherikh, Andres Gomez-Caminero.

**Writing – review & editing:** Wida S. Cherikh, Tzuyung Douglas Kou, Julia Foutz, Timothy J. Baker, Andres Gomez-Caminero.

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
