## [Decision Letter · Decision Letter 0]

18 Mar 2024

PONE-D-23-42267Patterns of belatacept use and risk of post-transplant lymphoproliferative disorder in US kidney transplant recipients: an analysis of the Organ Procurement and Transplantation Network databasePLOS ONE

Dear Dr. Gomez-Caminero,

Thank you for submitting your manuscript to PLOS ONE. After careful consideration, we feel that it has merit but does not fully meet PLOS ONE’s publication criteria as it currently stands. Therefore, we invite you to submit a revised version of the manuscript that addresses the points raised during the review process.

We look forward to receiving your revised manuscript.

Kind regards,

Justyna Gołębiewska

Academic Editor

PLOS ONE

Journal Requirements:

"We would like to acknowledge Samantha Noreen, PhD, of United Network for Organ Sharing for 

her assistance with PSM analyses. Medical writing and editorial assistance were provided by 

Vasupradha Vethantham, PhD, Kakoli Parai, PhD, and Jennifer Robertson, PhD, of Ashfield 

MedComms, an Inizio company, and were funded by Bristol Myers Squibb."

"This study was supported by Bristol Myers Squibb (https://www.bms.com; Princeton, NJ, USA). The sponsor participated in the study design, data collection and analysis, decision to publish, and preparation of the manuscript."

"I have read the journal's policy and the authors of this manuscript have the following competing interests: W.S.C., J.F., and T.J.B. have served as consultants to and have received research support from Bristol Myers Squibb. T.D.K. was a salaried employee of Bristol Myers Squibb and is currently an employee of BeiGene USA, Inc. A.G.-C. was a salaried employee of Bristol Myers Squibb and is currently an employee of Merck & Co., Inc." 

**Additional Editor Comments:**

**ACADEMIC EDITOR:**Please revise the manuscript according to the Reviewers' comments.

Reviewers' comments:

Reviewer's Responses to Questions

**Comments to the Author**

1. Is the manuscript technically sound, and do the data support the conclusions?

Reviewer #1: Yes

Reviewer #2: No

2. Has the statistical analysis been performed appropriately and rigorously? 

Reviewer #1: Yes

Reviewer #2: I Don't Know

3. Have the authors made all data underlying the findings in their manuscript fully available?

Reviewer #1: Yes

Reviewer #2: No

4. Is the manuscript presented in an intelligible fashion and written in standard English?

Reviewer #1: Yes

Reviewer #2: Yes

5. Review Comments to the Author

Reviewer #1: Thank you for allowing me to review this manuscript for your journal. The authors present a long awaited analysis of the incidence of PTLD in patients receiving belatacept versus those receiving a CNI. The authors did a wonderful job with the statistics and reporting and should be commended. I have a few comments to strengthen the manuscript:

1. The authors need to make it more clear how they "know" patients are on belatacept or CNI throughout the 5 year period. In the methods, it is indicated that the TRR and TRF forms are utilized. These forms are routinely inaccurate. It would be an important addition to the manuscript to describe this in more detail. The results hinge on the confidence level of the data included. I am concerned that some patients started belatacept and stopped early and/or others started and stopped CNI at unknown time periods making the inclusion into different groups problematic.

2. The authors should include how long patients were on each therapy if known.

3. There may be a typo on page 13, "PTLD histopathology was reported in 8.9% of belatacept-treated...". Should this instead be 89%?

4. The paragraph on page 9 describing the baseline characteristics is somewhat clunky and not clearly written. It might be better instead to just refer the reader to table 1 and avoid redundancy.

Reviewer #2: This study attempted to compare the rates of PTLD in adult EBV +ve kidney transplant recipients who have received belatacept vs CNI between Jun 15 2011 to Jun 14 2016 using OPTN database. It's difficult to draw strong conclusions given the confounding variables, the small # of events in both groups and short duration of follow-up.

The following confounders will increase the risk of PTLD and would suggest including some discussions in the limitations:

In the overall cohort,

- The induction agents used were very different between the two groups favoring belatacept: Antithymocyte globulin (ATG) was used in 50.8% vs 22.9% and campath 15.7% vs 9.5% while basiliximab 59.5% vs 18.7% in CNI vs belatacept group, respectively (Table 1)

- Mean/median doses of ATG and Campath used in each group, if available

- rejection rate at 1 year 22.3% vs 9.5% in belatacept vs CNI group - while it's true that maintenance immunosuppressants would usually be intensified, this is rejection rate at 1 year and it's unclear of rejection rates after 1 year. Also, if pts were switched from CNI to belatacept, the risk of rejection would be higher during the switchover period (this may also explain why half of the belatacept pts were also taking CNI at the same time)

- What were the rejection rates during the entire study period? Patients were recruited between 2011 to 2015 and followed until 2019.

- half of the belatacept group was also taking CNI with 3 PTLD cases in this combo group. I assume the combination group was included in the belatacept group to increase its sample size. Were these patients being transitioned off CNI ?

- duration of follow-up in each group was unclear

Table 2 - were all pts followed for 5 years if last patient was enrolled in Jun 2016 and study closed in Apr 2019. What was the mean/median duration of follow-up in each group?

- while most PTLD cases surfaced early post-transplant, the LONG TERM PTLD risk of belatacept is unknown as BENEFIT and BENEFIT-EXT were relatively short duration. It would be more interesting to follow these patients for a longer period of time to realize the true PTLD risk of belatacept

Table 3 PTLD cases - 47.6% of CNI group received other/multiple drugs vs 0% in the belatacept group. What were these agents and could these have influenced the PTLD risks

S1. Why are ATG & 1-yr rejection rates not included in the model? Of the covariates used for matching, these are likely more relevant to the outcome of interest. Please explain 'region observations', 'all observations'

S2 Fig is difficult to understand. Please label the y-axis and legend accordingly. please explain 'region', 'observations', Control=CNI, treated = belatacept

Event rates were low in both overall and PSM cohorts - this could explain why no statistically significant differences were found between belatacept and CNI groups. I'm not sure if we can conclude 'the risk of PTLD is comparable to that observed among pts receiving CNI-based immunosuppression' (p.19)

Discussions - seems biased to me just focusing on confounders that biased against belatacept (p.17-18); underreporting of PTLD in CNI treated patients (p.19). Were all PTLD cases reported in belatacept group and only CNI group was underreported?

6. PLOS authors have the option to publish the peer review history of their article (what does this mean?). If published, this will include your full peer review and any attached files.

Reviewer #1: No

Reviewer #2: No

---

## [Author Response · Author response to Decision Letter 0]

21 Jun 2024

Emily Chenette, PhD

Editor-in-Chief

PLOS ONE

June 21, 2024

Re: PONE-D-23-42267

Dear Dr. Chenette,

On behalf of my coauthors, I would like to thank you and the reviewers for considering our manuscript entitled, “Patterns of belatacept use and risk of post-transplant lymphoproliferative disorder in US kidney transplant recipients: an analysis of the Organ Procurement and Transplantation Network database” for PLOS ONE. We are pleased to submit both clean and tracked versions of the revised manuscript, along with a point-by-point response to the reviewers’ comments (below). Please note that line and page numbers in the point-by-point response refer to the highlighted manuscript with revisions shown inline. 

In addition, we confirm that we have addressed any issues raised by the Editorial Office. We have revised our Funding statement as follows: “This study was supported by Bristol Myers Squibb (https://www.bms.com; Princeton, NJ, USA). The sponsor participated in the study design, data curation and analysis, decision to publish, and preparation of the manuscript; and funded medical writing and editorial assistance for development of the manuscript.” The Acknowledgments have been edited as follows: “We would like to acknowledge Samantha Noreen, PhD, of United Network for Organ Sharing for her assistance with PSM analyses. Medical writing and editorial assistance were provided by Vasupradha Vethantham, PhD, Kakoli Parai, PhD, and Jennifer Robertson, PhD, of Ashfield MedComms, an Inizio company.” We have also updated the Competing Interests statement as follows: “W.S.C., J.F., and T.J.B. have served as consultants to and have received research support from Bristol Myers Squibb. T.D.K. was a salaried employee of Bristol Myers Squibb at the time of the study and is currently an employee of Daiichi Sankyo US. A.G.-C. was a salaried employee of Bristol Myers Squibb at the time of the study and is currently an employee of Merck & Co., Inc. This does not alter our adherence to PLOS ONE policies on sharing data and materials.”

Once again, thank you for considering our manuscript for publication. We look forward to your response. Please do not hesitate to contact me with any questions.

Sincerely,

Andres Gomez-Caminero

Merck & Co., Inc., Rahway, NJ, USA

Email: Agc11413@yahoo.com

The response has been uploaded as "Cover Letter" file.

---

## [Decision Letter · Decision Letter 1]

27 Sep 2024

Patterns of belatacept use and risk of post-transplant lymphoproliferative disorder in US kidney transplant recipients: an analysis of the Organ Procurement and Transplantation Network database

PONE-D-23-42267R1

Dear Dr. Gomez-Caminero,

We’re pleased to inform you that your manuscript has been judged scientifically suitable for publication and will be formally accepted for publication once it meets all outstanding technical requirements.

Kind regards,

Catherine A. Brissette, Ph.D.

Academic Editor

PLOS ONE

Additional Editor Comments (optional):

Reviewers' comments:

Reviewer's Responses to Questions

**Comments to the Author**

1. If the authors have adequately addressed your comments raised in a previous round of review and you feel that this manuscript is now acceptable for publication, you may indicate that here to bypass the “Comments to the Author” section, enter your conflict of interest statement in the “Confidential to Editor” section, and submit your "Accept" recommendation.

Reviewer #2: All comments have been addressed

Reviewer #3: All comments have been addressed

2. Is the manuscript technically sound, and do the data support the conclusions?

Reviewer #2: Yes

Reviewer #3: Yes

3. Has the statistical analysis been performed appropriately and rigorously? 

Reviewer #2: Yes

Reviewer #3: Yes

4. Have the authors made all data underlying the findings in their manuscript fully available?

Reviewer #2: Yes

Reviewer #3: Yes

5. Is the manuscript presented in an intelligible fashion and written in standard English?

Reviewer #2: Yes

Reviewer #3: Yes

6. Review Comments to the Author

Reviewer #2: Thank-you for the revisions and addressing my questions/comments!

I do not any further suggestions.

Reviewer #3: In my opinion, all the reviewers' comments are properly addressed. I only suggest considering in the discussion that low-dose CNI could be combined with belatacept to avoid the risk of acute rejection (and additional multiple treatments) with good outcomes also on the risk of PTLD (10.1371/journal.pone.0240335).

7. PLOS authors have the option to publish the peer review history of their article (what does this mean?). If published, this will include your full peer review and any attached files.

Reviewer #2: No

Reviewer #3: No

---

## [Editor Report · Acceptance letter]

11 Nov 2024

PONE-D-23-42267R1 

PLOS ONE

Dear Dr. Gomez-Caminero, 

I'm pleased to inform you that your manuscript has been deemed suitable for publication in PLOS ONE. Congratulations! Your manuscript is now being handed over to our production team.

Kind regards, 

on behalf of

Dr. Catherine A. Brissette 

Academic Editor

PLOS ONE